# Retrieval of Ocean Wind Speed Using Super-Resolution Delay-Doppler Maps

Hao-Yu Wang and Jyh-Ching Juang *

Department of Electrical Engineering, National Cheng Kung University, Tainan City 701, Taiwan;
n28044020@gs.ncku.edu.tw
* Correspondence: juang@mail.ncku.edu.tw

**Abstract:** The use of reflected Global Navigation Satellite System (GNSS) signals has shown to be effective for some remote sensing applications. In a GNSS Reflectometry (GNSS-R) system, a set of delay-Doppler maps (DDMs) related to scattered GNSS signals is formed and serves as a measurement of ocean wind speed and roughness. The design of the DDM receiver involves a trade-off between computation/communication complexity and the effectiveness of data retrieval. A fine-resolution DDM reveals more information in data retrieval while consuming more resources in terms of onboard processing and downlinking. As a result, existing missions typically use a compressed or low-resolution DDM as a data product, and a high-resolution DDM is processed for special purposes such as calibration. In this paper, a deep learning, super resolution algorithm is developed to construct a high-resolution DDM based on a low-resolution DDM. This may potentially enhance the data retrieval results with no impact on the instrument design. The proposed method is applied to process the DDM products disseminated by the Cyclone GNSS (CYGNSS) and the effectiveness of wind speed retrieval is demonstrated.

**Keywords:** Global Navigation Satellite System Reflectometry (GNSS-R); delay-Doppler map (DDM); single image super-resolution (SISR); very-deep super-resolution (VDSR); wind speed retrieval

## 1. Introduction

Global Navigation Satellite System Reflectometry (GNSS-R) is a bistatic remote sensing technique in which the GNSS-R receiver acquires and processes scattered signals from the Earth's surface that are generated by GNSS satellites. The GNSS-R technique can be utilized to retrieve geophysical parameters such as sea surface height [1,2], ocean surface wind [3,4], vegetation [5,6], and soil moisture [7,8], mainly based on delay-Doppler maps (DDMs). It was noted that several studies have been conducted on GNSS-R with phase observation (i.e., carrier phase measurement) [9–12], which is not included in DDMs and is not discussed in this paper. To generate a DDM, the GNSS-R receiver performs a series of correlation processing procedures at different code phase delays and Doppler frequency bins. Since scattered signals are relatively weak, a one-second long integration period is required, especially for spaceborne missions [13–15]. Currently, the National Aeronautics and Space Administration (NASA) Cyclone GNSS (CYGNSS) mission collects global surface wind field information to help predict the dynamics of hurricanes [16]. The CYGNSS is composed of eight micro-satellites. Each satellite uses four parallel channels to generate fully resolved DDMs, which contain the scattering properties above the ocean, from four Global Positioning System (GPS) satellites. The fully resolved DDM is then compressed by the onboard compression algorithm to reduce the data transfer requirements. Therefore, the primary data for downlink and wind speed estimation are principally the compressed DDM. It should also be noted that the compression is not a lossless algorithm.

In the past, a few studies were conducted on reconstruction of the normalized radar cross-section (NRCS) filed from the GNSS-R DDM. Schiavulli and colleagues [17] proposed an image reconstruction method based on truncated singular value decomposition (TSVD) to reconstruct the NRCS-image (X-Y domain) from the DDM (DD domain). The simulation results showed that the nonhomogeneous area, a simulated region, could be easily recognized despite its imperfect size and shape. In a different study by Schiavulli and co-workers [18], the same reconstruction algorithm was performed on real GNSS-R DDMs from TechDemoSat-1 (TDS-1). The tested results demonstrated that the land/sea transitions could be well located from the reconstructed NRCS-image. Both of these studies focused on the identification of reflection surface transition, and the application of this technique could be oil pollution or sea ice detection. Nevertheless, the above research is not directly related to this study. The purpose of the discussion here is to clarify the difference between those studies and our work.

This paper aimed to investigate the potential alternative in GNSS-R data processing to balance the complexity of GNSS-R operations with data retrieval quality. Briefly, the proposed method involves applying a super-resolution algorithm to a DDM with lower resolution than usual and to then use the resulting super-resolution DDM for data retrieval. The algorithm is developed through a deep learning procedure based on previously collected data. To verify the proposed method, the low-resolution DDM is generated using real DDM data disseminated by the CYGNSS, which is viewed as the output obtained from a hypothetical GNSS-R receiver. Then, the low-resolution DDM is converted to a super-resolution DDM via a single-image super-resolution algorithm. Eventually, the observables derived from the super-resolution DDM are used to retrieve the ocean surface wind speed. We collected actual CYGNSS data recorded in 2018 to evaluate the feasibility of the proposed method. The experimental results demonstrated the effectiveness of the ocean surface wind speed retrieval. Through a numerical analysis, the results showed that the proposed method saved 94% of the data on DDM generation but resulted in only a 4% performance degradation in terms of the wind retrieval. Furthermore, the proposed method reduced the data volume for downlink by approximately 15%.

## 2. Materials and Methods

The proposed method in this paper was intended to recover DDMs from smaller sizes to larger sizes where the reconstructed DDM provides nearly identical wind speed performance to a normal DDM. The normal DDM (i.e., fully resolved DDM) is real, accessible spaceborne science data recorded from an existing space mission. The reconstructed DDM (i.e., super-resolution DDM), which is expected to be similar to normal DDM, is produced by the proposed method. A simple but straightforward wind speed retrieval method is also described in this section. In the results section, the retrieval method is used to evaluate the wind speed retrieval performance using the normal DDM and recovered DDM.

### 2.1. DDM Data Description

The original product generated by CYGNSS receivers was the delay-Doppler map (DDM). The DDM is processed with different processing procedures to produce four levels of data products. In this research, we used the Level 0 (L0) data product, version 2.1, which consists of fully resolved DDMs (FR-DDM) and compressed DDMs (C-DDM). The FR-DDM is a raw DDM in units of "counts" related to the total power processed by the DDM instrument (DDMI). In addition, the FR-DDM is made at a resolution of 20 Doppler by 128 delay bins, and the delay and Doppler resolution are, respectively, 0.25 chips and 500 Hz. The C-DDM is generated from the FR-DDM on the satellite using the specified compression algorithm [19]. The dimension of the C-DDM is thus converted into 11 Doppler × 17 delay bins, but the delay and Doppler resolution of the C-DDM are identical to the FR-DDM. For these two data products, the FR-DDM data are included in the NetCDF files under limited access through the CYGNSS SFTP server, while the C-DDM data are also contained in the NetCDF files along with a large amount of metadata (e.g., incidence angle, azimuth angle, antenna gain, etc.) and are available through the NASA Physical Oceanography Data Active Archive Center (PO.DAAC). The

NASA CYGNSS mission will further calibrate the C-DDM data to generate a Level 1 data product for the purpose of estimating the ocean surface wind speed.

In this research, we generated another DDM, called low-resolution DDM (LR-DDM), by downsampling the FR-DDM. The sample points are every eight chips along the delay axis and every four frequency bins along the Doppler axis, as shown in Figure 1. Therefore, the size of the LR-DDM is 10 Doppler × 16 delay bins in which the delay and Doppler resolution are, respectively, two chips and 1000 Hz. We assume that the LR-DDM is output from a fictitious DDMI. Subsequently, the LR-DDM is reconstructed to produce a super-resolution DDM (SR-DDM) using the proposed method, as described in Section 2.3, which is the final product used to retrieve wind speed. Table 1 shows the differences in the characteristics of the four types of DDMs described above, and Figure 2 shows an example pair of DDMs. Since the ideal SR-DDM is assumed to be identical to the FR-DDM, we only show three out of four DDMs here.

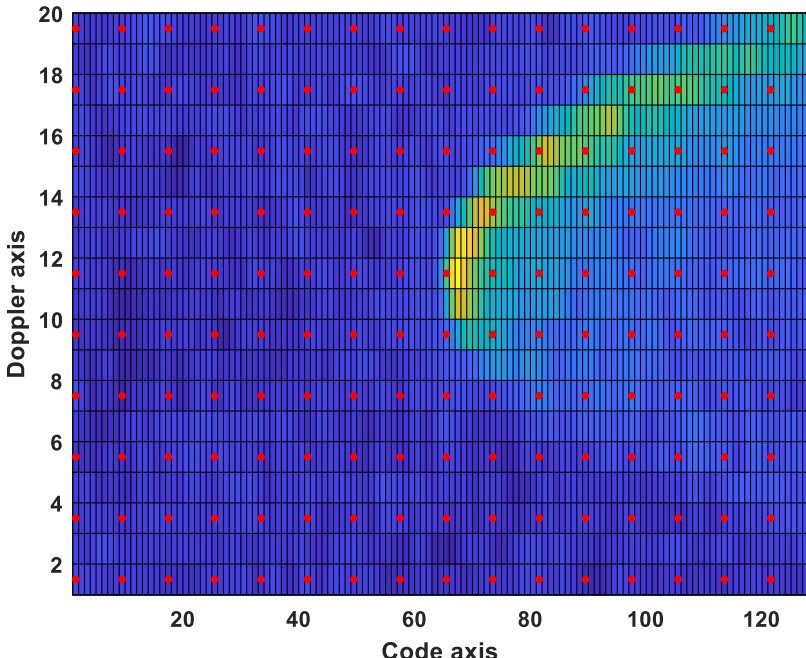

**Figure 1.** The sample points on the fully resolved delay-Doppler map (FR-DDM) used to produce the low-resolution DDM (LR-DDM). The red dot indicates the sample points to form the LR-DDM. The black rectangle represents the grid of the FR-DDM.

**Table 1.** Summary of delay-Doppler map (DDM) characteristics.

| Parameter | FR-DDM | C-DDM | LR-DDM | SR-DDM |
|---|---|---|---|---|
| Dimension | 20 × 128 | 11 × 17 | 10 × 16 | 20 × 128 |
| Doppler resolution | 500 Hz | 500 Hz | 1000 Hz | 500 Hz |
| Delay resolution | 0.25 chip | 0.25 chip | 2 chip | 0.25 chip |
| Source | CYGNSS DDMI | FR-DDM | FR-DDM | LR-DDM |

Abbreviations: FR, fully resolved; C, compressed; LR, low-resolution; SR, super-resolution; CYGNSS, cyclone Global Navigation Satellite System.; DDMI: DDM instrument.

Notably, unlike C-DDM data, which were the daily routine product for downlink, FR-DDM data were only recorded and downlinked relying on the DDMI operational mode given by the ground station command. We collected all of the FR-DDM data recorded in 2018 from the CYGNSS server. Table 2 shows the collected data set, including the recording date, source satellite number, and FR-DDM counts. Since the output rate of the FR-DDM is 1 Hz per channel, the recording period for each data set varies from a few minutes to several hours. It should be emphasized that the L0 recording mode is not

a common mode for the CYGNSS mission, and we downloaded all the available L0 data recorded in 2018. These 18 data sets were then combined and split into a training set and test set, which will be described in detail in the following section.

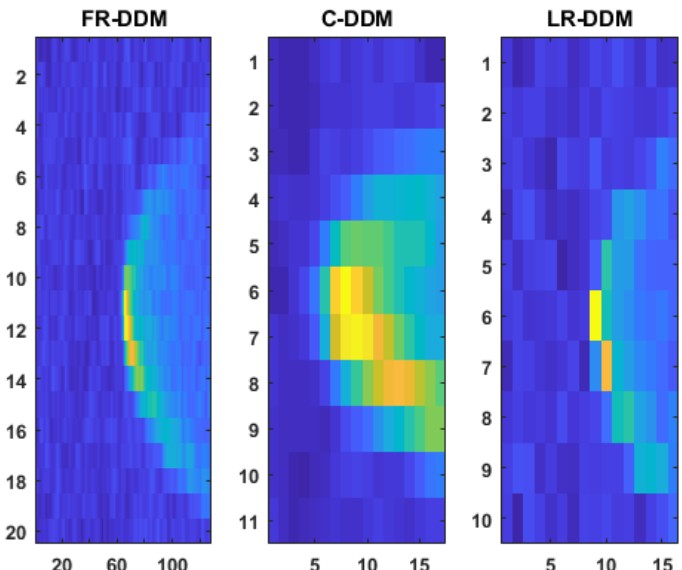

**Figure 2.** Example of CYGNSS DDM instrument (DDMI) onboard-processed 1 Hz FR-DDM (**left**), compressed DDM (**middle**), and author-generated LR-DDM (**right**).

**Table 2.** List of collected Level 0 (L0) data.

| No. | Date (Year/DOY) | Satellite id | FR-DDM Counts |
|---|---|---|---|
| # 1 | 2018/108 | 3 | 406 |
| # 2 | 2018/109 | 3 | 367 |
| # 3 | 2018/121 | 3 | 545 |
| # 4 | 2018/122 | 3 | 473 |
| # 5 | 2018/148 | 3 | 550 |
| # 6 | 2018/149 | 3 | 537 |
| # 7 | 2018/158 | 8 | 74294 |
| # 8 | 2018/159 | 8 | 38643 |
| # 9 | 2018/164 | 3 | 583 |
| # 10 | 2018/166 | 3 | 499 |
| # 11 | 2018/218 | 8 | 1613 |
| # 12 | 2018/219 | 2, 7 | 3869 |
| # 13 | 2018/220 | 3, 4 | 3479 |
| # 14 | 2018/254 | 1, 2, 3, 5, 8 | 4869 |
| # 15 | 2018/255 | 1, 2, 3, 4, 5, 6 | 6493 |
| # 16 | 2018/256 | 1, 3, 4, 5, 6, 7, 8 | 7416 |
| # 17 | 2018/257 | 1, 3, 5, 7, 8 | 5624 |
| # 18 | 2018/258 | 2, 3, 4, 5, 6, 8 | 6383 |

*2.2. Dataset of European Centre for Medium-Range Weather Forecasts (ECMWF)*

Matchup 10-m-referenced ocean surface wind speed data provided by the ECMWF dataset was used to develop an empirical geophysical model function (GMF) to evaluate the wind speed retrieval performance using the FR-DDM and SR-DDM. As an independent organization, the ECMWF dataset can be used to produce weather forecast services and climate reanalysis products. The horizontal resolution and temporal resolution of reanalysis products are $0.25° \times 0.25°$ and 1 h, respectively. As a result, bicubic interpolation in space and time is performed to estimate the ground truth wind speed at the locations and time of the CYGNSS reported specular point in the metadata. The reanalysis products

obtained from the ECMWF dataset are made available to the public through the Climate Data Store (CDS). Figure 3a shows the geographical distribution of the CYGNSS FR-DDM data acquired in 2018, and the color represents the count per point. Figure 3b shows the interpolated ECMWF wind speed at the times and locations of points in Figure 3a. The color in Figure 3b represents the interpolated wind measurements processed from the ECMWF reanalysis products. It is worth noting that approximately 150,000 of the collected data, the sum of all the points within the high count region, were highly concentrated off the coast of the southeastern United States. A comparison with the geographical distribution of wind speed measurements suggests that it is likely that the FR-DDM recording mode operated to collect high wind data. It should also be noted that data numbers #7 and #8 (refer to Table 2) account for 70% of the entire data set.

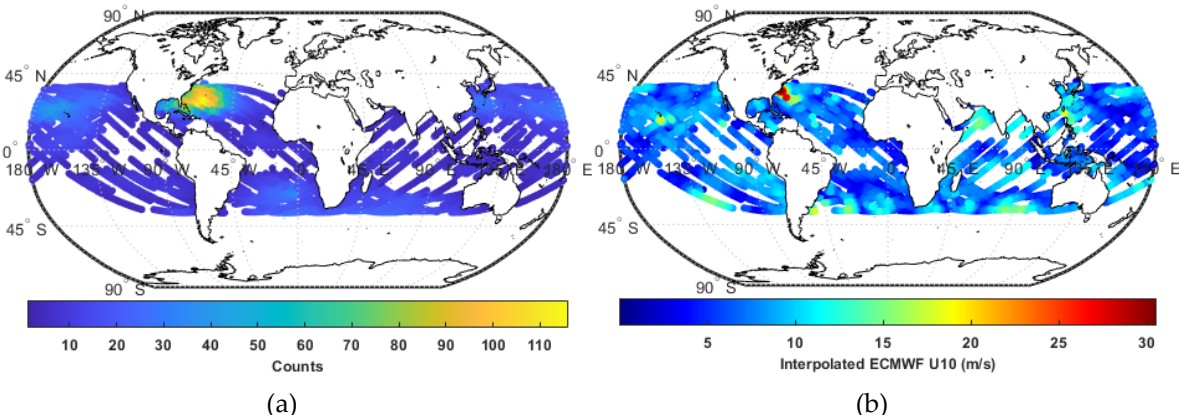

(a)  (b)

**Figure 3.** Geographical distribution of collected CYGNSS FR-DDM data in 2018 and corresponding interpolated European Centre for Medium-Range Weather Forecasts (ECMWF) 10-m-referenced ocean surface wind speed. (**a**) The color at the left part of the figure indicates the count per point. (**b**) The color at the right part of the figure indicates the coincident wind speed obtained by interpolating the ECMWF data.

### 2.3. Super-resolution DDM Reconstruction Model

The single image super-resolution (SISR) algorithm was aimed toward solving the problem related to recovering the high-resolution (HR) image from a low-resolution (LR) image, as depicted in Figure 4. To date, many methods have been proposed to solve this problem [20–22]. Among these methods, very-deep super-resolution (VDSR) proposed by Kim and colleagues in 2016 is the first very deep model to perform SISR [23]. Compared with traditional SISR methods, the VDSR network has made two contributions, which are also why we chose this method. The first is that the reconstructed image using the VDSR network provides peak signal-to-noise ratio, a metric to measure reconstruction quality of image, at least more than 0.82 dB than the previous method. The second contribution is that the inference time, the time to complete processing from an input image to output image, of the VDSR network is up to 10 times faster than the conventional method. The primary characteristics of the VDSR are as follows: (1) The VDSR is made up of a 20 layer VGG-net; (2) it can be trained with a relatively high initial learning rate that results in faster convergence speed; (3) it supports multiple scales with a single model; (4) it learns the residual between an HR image and LR image, which can lead to improved performance and accelerated convergence. In this research, we considered the DDM as an image, and planned to modify and train the VDSR network using the prepared DDM set.

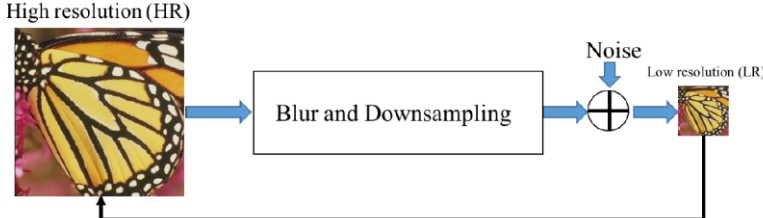

**Figure 4.** Block diagram of the single image super-resolution (SISR) problem [22].

Since VDSR is a residual learning model, it is necessary to conduct pre-processing on the DDM before training the model. The VDSR learns the residual image obtained by subtracting the LR image (i.e., the LR-DDM) that has been enlarged using bicubic interpolation from the reference HR image (i.e., the FR-DDM). The high-frequency image information is incorporated into a residual image. Generally, an image will be converted from an RGB color space to a luminance and chrominance color space that represents the brightness and color-difference information for each pixel, and the VDSR uses only the luminance channel for training. As DDM is not a color image, we converted it to grayscale space instead. Suppose $I_{FR-DDM}$ denotes the grayscale of the fully-resolved DDM (HR image), and $I_{LR-DDM}$ denotes the upscaled downsampled DDM (LR image). Then, the VDSR network uses $I_{LR-DDM}$ as input and learns to estimate $\hat{I}_{res} = I_{FR-DDM} - I_{LR-DDM}$ from the training data. In summary, the VDSR network takes the upscaled LR-DDM as input and predicts the residual corresponding to its original FR-DDM counterpart. After the VDSR network learns to estimate the DDM residual, the SR-DDM is reconstructed by adding the estimated residual image to the upscaled LR-DDM and converting the grayscale value back to the "counts" value.

The VDSR network consists of 20 layers. Except for the first layer and the last layer, the middle 18 layers are of the same type, where each is a convolutional layer containing 64 $3 \times 3 \times 64$ (height × width × depth) filters. Every convolutional layer is followed by a rectified linear unit (ReLU) layer. The first layer works on an input image with a specific patch size. The patch size was set at $41 \times 41$ in the original research because this is the ideal receptive field for a network with a 20-later depth [23]. In our case, however, this size exceeds the dimension of the input DDM in the Doppler axis. Therefore, we considered two schemes in this research: (1) Set the patch size at $10 \times 16$; (2) replace the middle 18 layers with seven layers (a total of nine layers) and set the patch size at $19 \times 19$. The last layer is a regression layer used for image reconstruction, which is a convolutional layer consisting of a single $3 \times 3 \times 64$ filter that is not followed by a ReLU layer. Figure 5 shows the structure of the VDSR network.

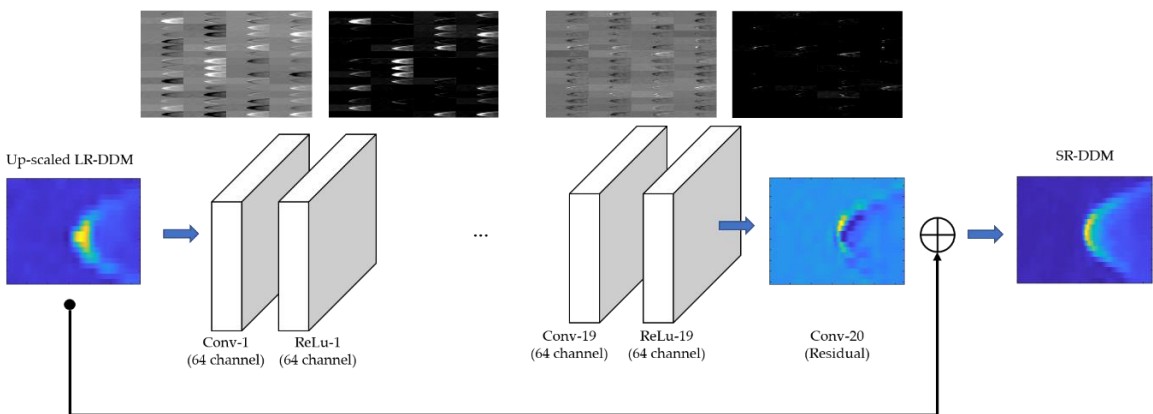

**Figure 5.** The structure of the very-deep super-resolution (VDSR) network.

The pre-processed data, the FR-DDM, LR-DDM, and corresponding DDM residual, are filtered before training the VDSR network. The data for which the DDM signal-to-noise ratio (SNR; reported

in CYGNSS metadata) was less than 3 dB are excluded. The effect of the 3 dB threshold was to remove some noise-only DDMs. The remaining data set was reduced to 60% of the original data collection set, but there were still 96,000 pairs of data that can be used for the analysis. We further split the data set into two independent sets: a randomly selected training set, which accounts for 70% of samples, and a remaining test set. These two data sets are also used in the discussion in the subsequent section.

### 2.4. Retrieval of Wind Speed and the Bistatic Radar Equation

Retrieval of the ocean surface wind speed is the most common application of GNSS-R remote sensing. To obtain the inverse wind speed, it is necessary to define the observables from the measured DDMs, regress them against the collocated wind speeds using other methods, and associate the observables with the wind speed through an empirical geophysical model function (GMF). CYGNSS has put a great deal of effort into the development of a wind speed retrieval algorithm [24,25]. However, their method requires suitable calibration on DDM products before extracting the usable observables [26,27]. In this paper, we applied a wind speed retrieval algorithm based on the average power of the DDM and the bistatic radar equation to evaluate the performance using the FR-DDM.

The bistatic radar equation describes the dependence between the DDM and ocean surface roughness; derived by Zavorotny and Voronovich in 2000 [28] it can be expressed as follows:

$$\left\langle \left| Y(\tau, f) \right|^2 \right\rangle = \frac{P_t G_t \lambda^2}{(4\pi)^3} T_i^2 \iint\limits_A \frac{G_r \chi^2(\tau, f)}{R_t^2 R_r^2} \sigma_0 dA , \tag{1}$$

where $\left\langle \left| Y(\tau, f) \right|^2 \right\rangle$ represents the correlation power between the received scattered signals and the local replicas at code phase delay $\tau$ and Doppler frequency $f$, which is also the bin value of the DDM; $P_t$ and $G_t$ are the GPS satellite transmitting power and antenna gain, respectively; $\lambda$ is the carrier wavelength of the L1 band; $T_i$ is the coherent integration time; $G_r$ is the gain of the nadir-looking antenna at the specular point; $R_t$ is the distance between the GPS transmitter and the specular point; $R_r$ is the distance between the GNSS-R receiver and the specular point; $\chi^2(\tau, f)$ is Woodward's ambiguity function (WAF); and $\sigma_0$ is the bistatic cross section coefficient. Among all these parameters, $\sigma_0$ reveals the variations in ocean surface roughness affected by different wind strength. Therefore, we calculate $\sigma_0$ as the observable for wind speed inversion using the DDM and other obtainable parameters. Given some simplifying assumptions, Equation (1) can be given as:

$$\langle \sigma_0 \rangle = \frac{(4\pi)^3 R_t^2 R_r^2}{P_t G_t \lambda^2 T_i^2 G_r A_0} \overline{P}_r, \tag{2}$$

where $\langle \sigma_0 \rangle$ is the average of $\sigma_0$ according to the received power $\overline{P}_r$, for simplified receiving area $A_0 = 1/\cos(\theta_{inc})$, where $\theta_{inc}$ is the incidence angle. The metadata in the CYGNSS product can be substituted into the equation as follows: *gps_tx_power_db_w* for $P_t$, *gps_ant_gain_db_i* for $G_t$, *sp_rx_gain* for $G_r$, *tx_to_sp_range* for $R_t$, *rx_to_sp_range* for $R_r$, and *sp_inc_angle* for $\theta_i$. The term $\overline{P}_r$ is the estimated received power of the DDM and is computed from the signal-to-noise (SNR) of the DDM based on [29] the following equation:

$$SNR = \frac{P_{avg} - \langle N \rangle}{\langle N \rangle}, \tag{3}$$

where $P_{avg}$ is the power value comprising the signal and noise power of the DDM and is calculated as the average value centered at the peak power across three Doppler bins (−1000 1000) and four delay bins (−0.25 0.25). $\langle N \rangle$ is the noise floor within the signal-free area of the DDM, where it starts from the first 45 delay bins across all Doppler bins. The size used to calculate $P_{avg}$ is equal to a spatial resolution of around 25 km and may be changed between 22 and 30 km depending on the incidence angle between the specular point and the GNSS-R receiver [30].

The retrieval model develops a GMF that relates the observable with a coincident wind speed obtained from ECMWF data. The relationship between the training set observable computed using Equation (3) and the collocated ECMWF wind speed is shown in Figure 6a. We simply relate the observables and the wind speed by fitting the following form:

$$U_{10} = Ae^{Bx} + C, \tag{4}$$

where $x$ refers to the observable computed using Equation (3). Using a non-linear least squares regression, parameters A, B, and C are respectively, $3.377 \times 10^{19}$, $-0.170$, and 0.350 for the FR-DDM training set. The retrieved wind speed obtained by applying the GMF to the test set is shown in Figure 6b. The retrieved wind speed reports a result with a bias of 0.159 m/s and a root-mean-square error (RMSE) of 2.064 m/s. The wind speed retrieval method described here is used in the results section to estimate the wind speed with the SR-DDM to validate its wind sensing performance.

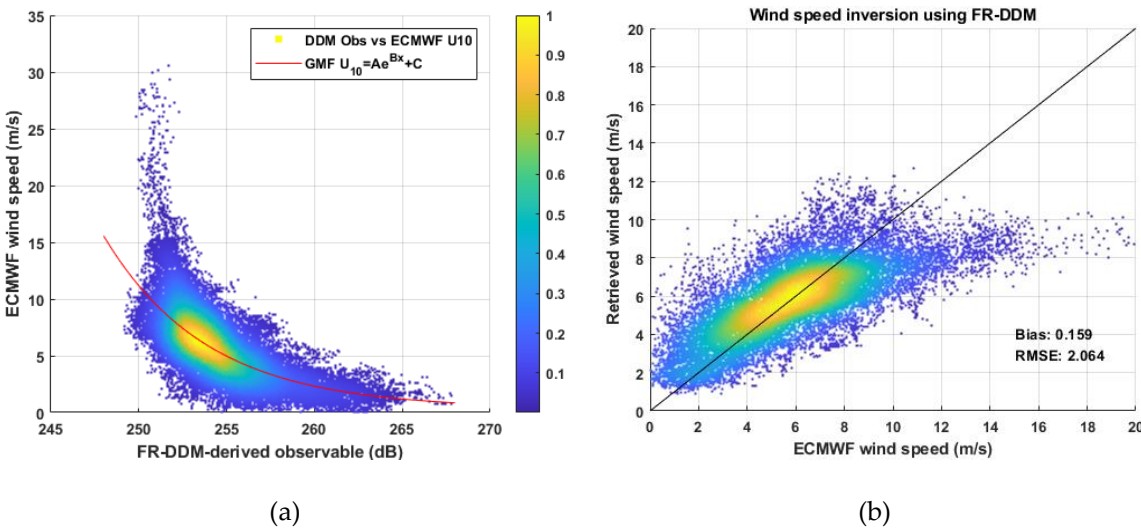

(a)                                                        (b)

**Figure 6.** (**a**) CYGNSS FR-DDM-derived $\sigma_0$ versus collocated ECMWF wind speed for the training data set with the fitted geophysical model function (GMF) (in red) of the form $U_{10} = Ae^{Bx} + C$; (**b**) CYGNSS FR-DDM retrieved wind speed with the fitted GMF versus collocated ECMWF wind speed for the test data set.

### 2.5. Summary of the Proposed Method

Figure 7 shows the overall flow diagram for this research. The methodology of the proposed method is composed of three parts. First, adequate CYGNSS FR-DDM data are chosen the generate the LR-DDM and train the DDM reconstruction network. Subsequently, the DDM reconstruction network is used to create the SR-DDM from the LR-DDM. Eventually, the SR-DDM is used to develop a specialized wind speed GMF. In the assessment section, the performance is evaluated by comparing the SR-DDM-derived wind speed with the FR-DDM-derived wind speed. It should be noted that the purpose of this paper is to propose an algorithm to derive high-resolution DDMs from downscaled DDMs. The wind speed retrieval method, which is not a state-of-art retrieval method, is used to verify the feasibility of the proposed method. The gray line shows a conventional way from data pre-processing, including preparation of ground truth and quality control, to the development of the wind retrieval function in Figure 7. The red line shows the procedure for implementing the proposed method in Figure 7. The green line, shown in Figure 7, indicates the retrieved wind speed derived from the FR-DDM and SR-DDM, respectively. The difference between the two estimated wind speeds is to evaluate the performance of the proposed method.

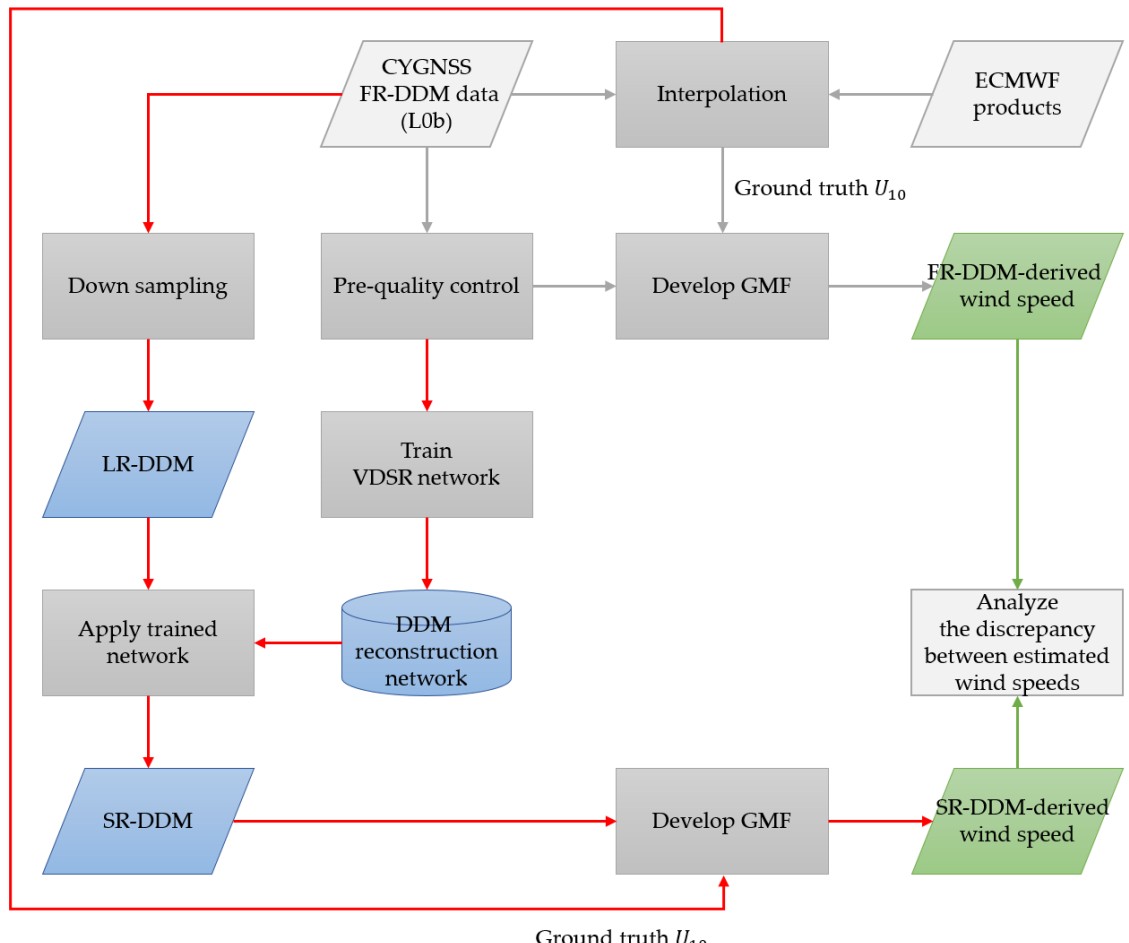

**Figure 7.** Overall research flow diagram. The line indicates the data flow in this research. The red line indicates the data flow of applying the proposed method from generating the SR-DDM to developing the corresponding wind GMF. The gray line indicates the data flow of using the FR-DDM to develop the corresponding wind GMF. The green line indicates the retrieved wind using the FR-DDM and SR-DDM, respectively.

## 3. Results

In this section, we first look at the quality of the SR-DDM from the perspective of image restoration. Due to the uneven distribution of data (the FR-DDM from satellite-8 dominates the whole data set), we observe the reconstruction situation using the FR-DDM made from different satellites. After that, the retrieved wind speed using the SR-DDM and FR-DDM are compared to evaluate the performance degradation resulting from reconstruction distortion. Finally, the benefits of the proposed method are provided.

### 3.1. Quality of Super-Resolution DDM

This section presents the performance evaluation of the SR-DDM from the image processing point of view. Therefore, we apply two standard metrics used in image restoration: the peak signal-to-noise ratio (PSNR) and the structural similarity index (SSIM) [21,22,31] where these two metrics are commonly used to measure the reconstruction quality of lossy transformation. The PSNR (in dB) computation is defined as [22]:

$$PSNR = 10 \times \log_{10}\left(\frac{P^2}{MSE}\right), \tag{5}$$

where $P$ is the maximum possible peak value based on the image datatype. MSE is the mean square error defined as:

$$MSE = \frac{1}{N}\|I - \hat{I}\|_F^2 \, , \tag{6}$$

where $I$ is the reference image, $\hat{I}$ is the reconstructed image both with $N$ pixels, and $\| \cdot \|_F^2$ is the Frobenius norm. The PSNR is used to measure the lossy transformation, where a higher value is better. On the other hand, the structural similarity index (SSIM) is defined as [22]:

$$SSIM(I, \hat{I}) = \frac{2\mu_I\mu_{\hat{I}} + k_1}{\mu_I^2 + \mu_{\hat{I}}^2 + k_1} \times \frac{\sigma_{\hat{I}I} + k_2}{\sigma_I^2 + \sigma_{\hat{I}}^2 + k_2} \, , \tag{7}$$

where $\mu_I$ and $\sigma_I^2$ denote the mean and variance of the reference image, $\sigma_{\hat{I}I}$ is the covariance between the reference image and reconstructed image, and $k_1$ and $k_2$ are constant relaxation terms. The SSIM is used to measure the structural similarities between the reference image and the reconstructed image, where the range is from 0 to 1.

Figure 8 shows the SR-DDM reconstructed from the LR-DDM using the model described in Section 2.3. Several examples were selected to display the reconstruction results. It is obvious that most of the information in the FR-DDM, especially the signal power distribution, can be recovered from the LR-DDM. Furthermore, the light SR-DDM (LSR-DDM), obtained by reconstructing the LR-DDM using the light model (LVDSR), provides very similar results as those for the SR-DDM.

It should be recalled that most of the FR-DDM data were generated by satellite-8, which means the model is mostly trained with the DDM coming from the same source. Thus, we investigated whether the DDM reconstruction model can recover the DDM generated by the other satellites. Figure 9 shows the image reconstruction metrics with different satellite numbers using the test data set. In the field of image restoration, typical values for the PSNR to measure the lossy image are between 30 and 50 dB [32]. As shown in Figure 9a, the results demonstrate that the SR-DDM of average quality can meet the conditions, and the performance is consistent across different satellites. Nevertheless, part of the PSNR is lower than the typical value. The reason for this is that the denoising effect of the VDSR smooths out not only power distribution but also the noise region of the DDM. As shown in the first, third, and fourth columns in Figure 8, it is obvious that the SR-DDM is relatively smooth compared to the FR-DDM. This effect will result in a larger MSE and thus, a smaller PSNR. Figure 9b displays the results from the SSIM metric, which agree well with the results shown in Figure 9a.

On the other hand, the reconstruction performance under coherent and incoherent scattering is also a matter of concern to the proposed method. It is known that the measured DDM is typically assumed under incoherent scattering over ocean surfaces [33]; that is, a 'horseshow' shape of the DDM, as shown in Figure 10a. However, the specular scattering can become coherent in some low wind speed conditions, resulting in a sharp shape of the DDM, as shown in Figure 10b. Figure 10 demonstrates an example of the FR-DDM under coherent and incoherent scattering with the corresponding SR-DDM reconstructed from the LR-DDM (not shown). Nevertheless, it is not simple to determine the coherent/incoherent component of a DDM, which is also beyond the scope of this research. An alternative way for verifying the reconstruction performance account for different scattering conditions is to analyze reconstruction metrics under different wind speeds and DDM SNRs. As shown in Figure 11, the proposed method shows that its reconstruction capacity is insensitive to the amount of data and wind speed. Most of the data fall in the wind speed range of 3–8 m/s, and the proposed method does not show overfitting, which is a common problem in deep learning-based approaches. In addition, the VDSR and LVDSR converge with similar performance. In Figure 12, it shows that the reconstruction performance grows with an increasing DDM SNR. As noted previously, the DDM with a lower SNR (i.e., the noisy DDM) is smoothed by the proposed method during reconstruction, resulting in lower PSNR and SSIM values. Further, LVDSR is slightly better but with a larger variance in this perspective. However, both of them

demonstrated remarkable reconstruction performance in any case. To sum up, the proposed method can produce SR-DDM under different scattering conditions.

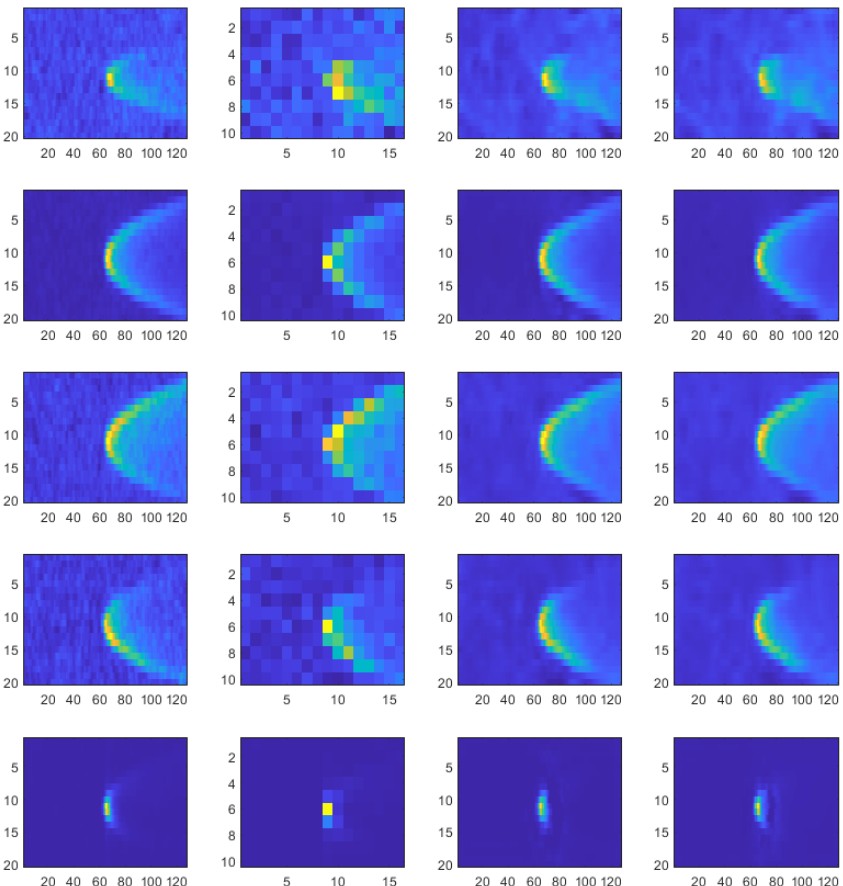

**Figure 8.** Examples of the DDM comparison of the FR-DDM, LR-DDM, and SR-DDM. The first and second columns are the FR-DDM and LR-DDM, respectively. The third and fourth columns are the SR-DDMs reconstructed by the VDSR in scheme 1 and scheme 2, respectively. The first row is the case of the lowest DDM signal-to-noise ratio (SNR) in the test set, and the fifth row is the case of the highest DDM SNR in the test set. The middle three rows are randomly selected cases.

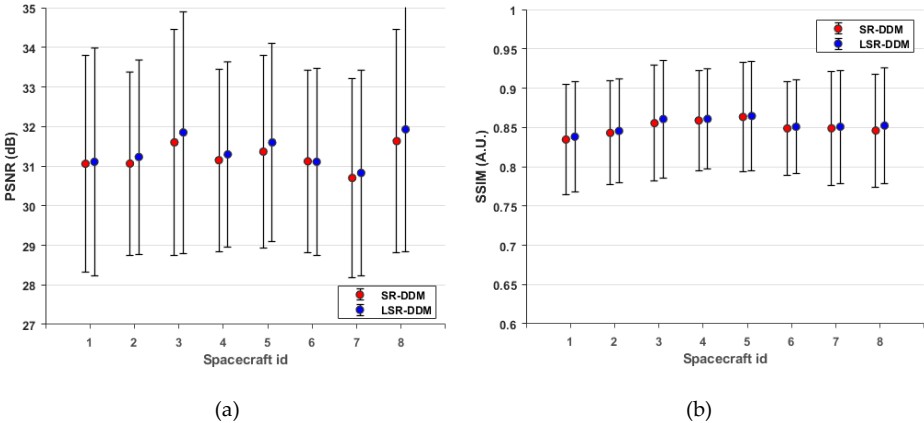

**Figure 9.** Quality of DDM reconstruction for different satellites. (**a**) Peak signal-to-noise ratio (PSNR) values versus satellite number for SR-DDM and LSR-DDM with one sigma variation. (**b**) Structural similarity index (SSIM) values versus satellite number for SR-DDM and LSR-DDM with one sigma variation.

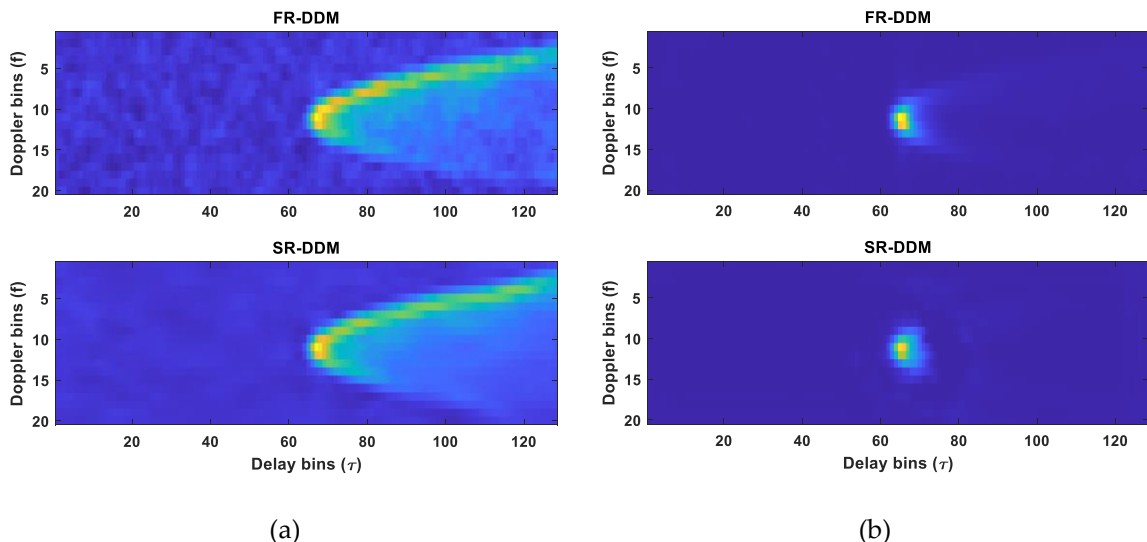

(a) (b)

**Figure 10.** Example of incoherent (**a**) and coherent (**b**) DDM in the test data set. The measured incoherent FR-DDM has an SNR of 3.80 dB with an ECMWF wind speed of 29.90 m/s. The measured coherent FR-DDM has an SNR of 20.68 dB with an ECMWF wind speed of 2.92 m/s. (a) The PSNR and SSIM for this SR-DDM are respectively 29.95 dB and 0.80. (b) The PSNR and SSIM for this SR-DDM are respectively 36.82 dB and 0.87.

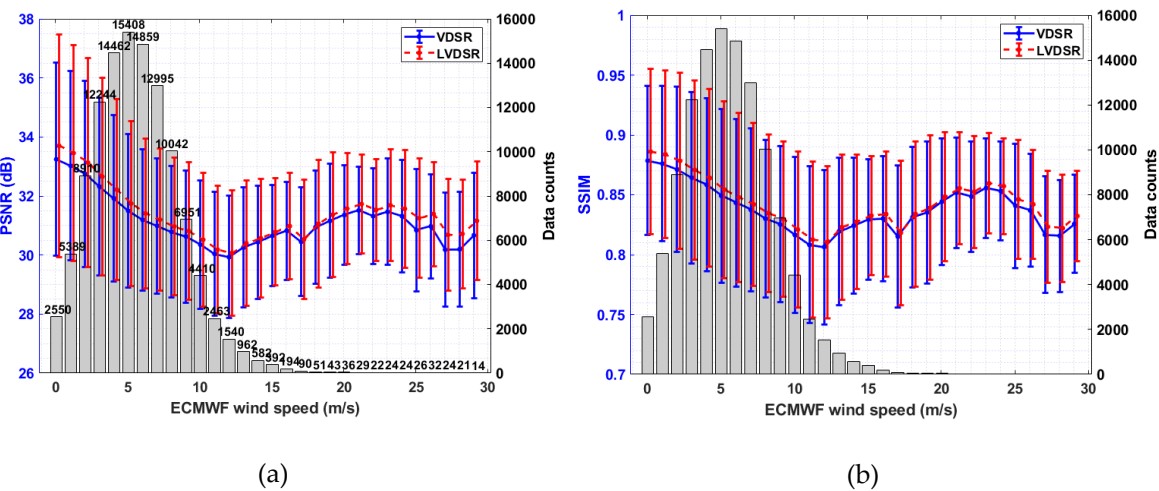

(a) (b)

**Figure 11.** Quality of DDM reconstruction for different wind speeds. (**a**) PSNR value versus ECMWF wind speed for VDSR (blue) and LVDSR (red), along with the data histogram in the background (right vertical axis). (**b**) SSIM value versus ECMWF wind speed for VDSR (blue) and LVDSR (red), along with the data histogram in the background (right vertical axis).

In this section, we investigate the reconstruction performance using two general metrics in different aspects. In summary, the results indicate that the proposed method can perform DDM reconstruction under any conditions. Nonetheless, these analyses were conducted from the image restoration point of view. Consequently, it is necessary to carry out further examinations to verify the feasibility of the proposed method for wind speed retrieval. This assessment will be discussed in detail in the subsequent section.

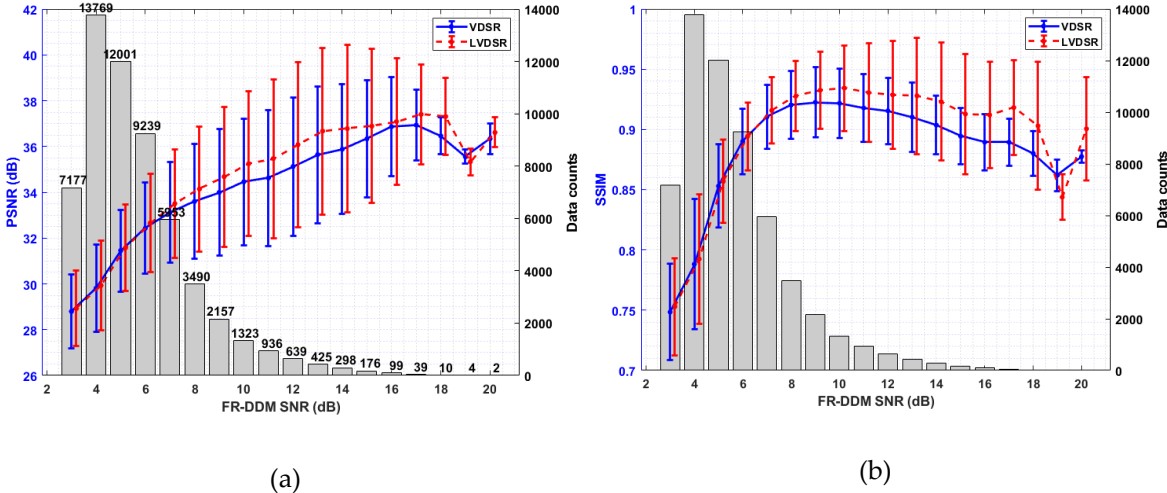

**Figure 12.** Quality of DDM reconstruction for different FR-DDM SNR. (**a**) PSNR value versus FR-DDM SNR for VDSR (blue) and LVDSR (red), along with the data histogram in the background (right vertical axis). (**b**) SSIM value versus FR-DDM SNR for VDSR (blue) and LVDSR (red), along with the data histogram in the background (right vertical axis).

### 3.2. Performance Evaluation of SR-DDM-derived Wind Speed

To evaluate the feasibility of retrieving wind speed using the SR-DDM, we followed the procedure in Section 2.4 to develop the independent GMF and analyze the estimated wind speed. There are two kinds of SR-DDMs. The first is the general SR-DDM reconstructed using the 20-layer VDSR network with a patch size of $10 \times 16$. The second is the SR-DDM reconstructed using the 10-layer VDSR network with a patch size of $19 \times 19$, denoted as the LSR-DDM. Scatter plots between calculated observables and each independent fitted wind speed GMF for the SR-DDM and LSR-DDM are shown in Figures 13 and 14, respectively. It should be noted that the scatter plots in Figures 13a and 14a use the same training data set while the scatter plots of Figures 13b and 14b use the same test data set. The only difference is the architecture of the DDM reconstruction network.

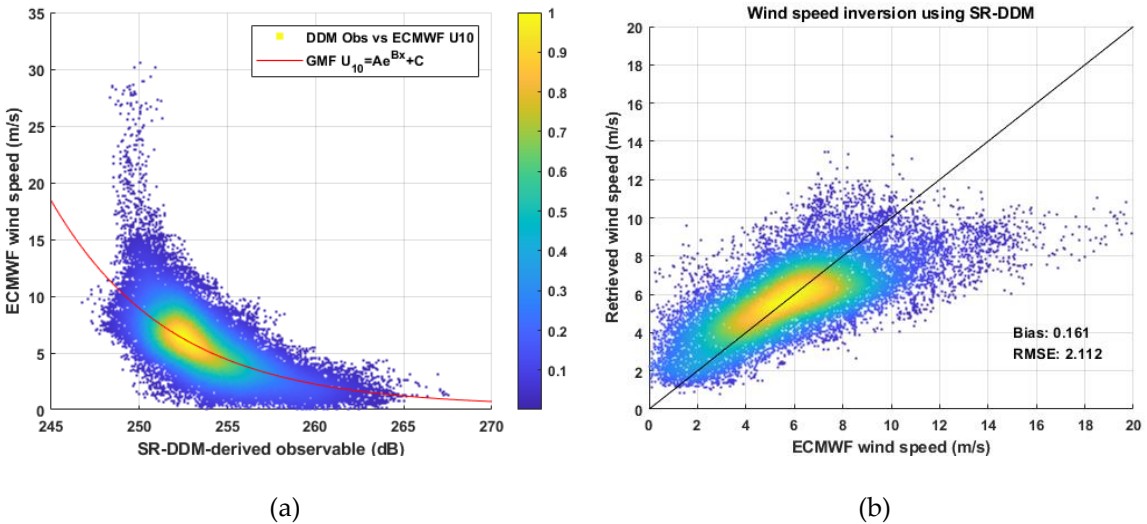

**Figure 13.** (**a**) SR-DDM-derived $\sigma_0$ versus interpolated ECMWF wind speed for the training data set with the fitted GMF (in red) of the form $U_{10} = Ae^{Bx} + C$; (**b**) SR-DDM retrieved with the fitted GMF versus collocated ECMWF wind speed for the test data set.

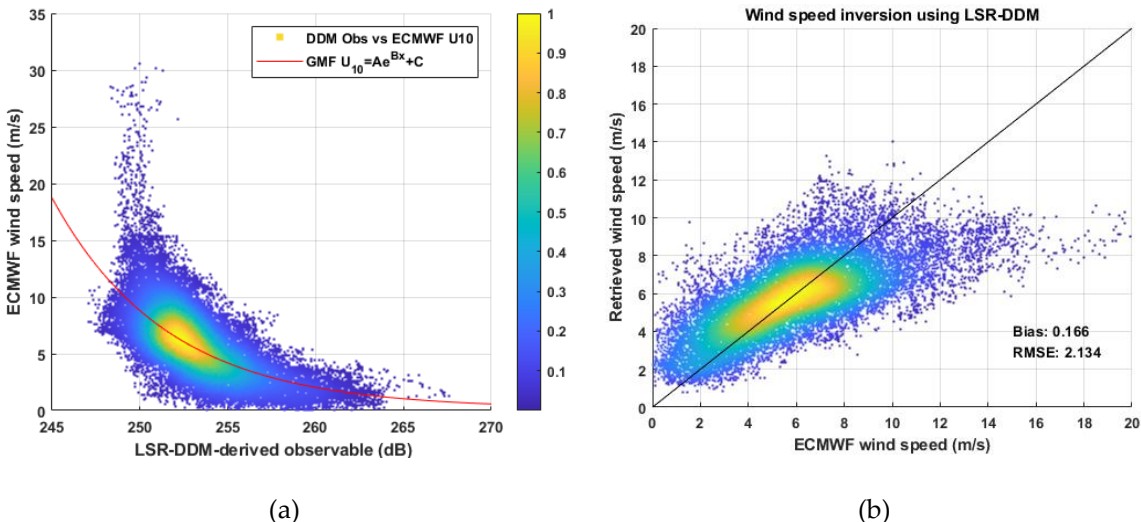

**Figure 14.** (**a**) LSR-DDM-derived $\sigma_0$ versus interpolated ECMWF wind speed for the training data set with the fitted GMF (in red) of the form $U_{10} = Ae^{Bx} + C$; (**b**) LSR-DDM retrieved with the fitted GMF versus collocated ECMWF wind speed for the test data set.

Both Figures 13 and 14 show that meaningful observables can be extracted from both the SR-DDM and LSR-DDM. The fitting results also demonstrate that the retrieved wind speed is highly consistent with the ground truth wind speed. A comparison of the wind speed retrieval performance is provided in Table 3. The statistical analysis indicates that the FR-DDM-derived wind speed reports the best performance, with a bias error of 0.159 m/s and an RMSE value of 2.064 m/s. Nevertheless, both the SR-DDM-retrieved wind speed and the LSR-DDM-retrieved wind speed can provide nearly equivalent results to that of the FR-DDM.

**Table 3.** $U_{10}$ retrieval algorithm performance statistics. Bias and root-mean-square errors (RMSEs) are expressed in m/s. Pearson's R is also provided.

| Observable Source | GMF Parameters (Training Set) | | | Retrieved $U_{10}$ Error (Test Set) | | |
|---|---|---|---|---|---|---|
| | **A** | **B** | **C** | **Bias** | **RMSE** | **R** |
| FR-DDM | $3.377 \times 10^{19}$ | −0.170 | 0.350 | 0.159 | 2.064 | 0.72 |
| SR-DDM | $5.185 \times 10^{17}$ | −0.155 | 0.228 | 0.161 | 2.112 | 0.70 |
| LSR-DDM | $3.575 \times 10^{17}$ | −0.153 | 0.172 | 0.166 | 2.134 | 0.70 |

The results presented in this section indicate that the retrieved wind speed using the SR-DDM-based data can provide as reliable of an estimation value as that obtained utilizing the FR-DDM data. However, both results show more significant bias under high wind conditions since we used a simple regression method to develop the GMF. This could be improved by taking different sea states into account to build separate GMFs. A possible way to solve this problem can follow the method proposed by Ruf and Balasubramaniam [24]. The authors developed wind GMF under two conditions. One is a fully developed seas (FDS) GMF for low-to-moderate wind speeds using the numerical weather prediction (NWP) model outputs, similar to the ECMWF in this paper, as ground truth. The other is a young seas/limited fetch (YSLF) GMF based on associating the observable with measurements by hurricane hunter aircraft, equipped with the stepped frequency microwave radiometer (SFMR), during flights through hurricanes. Another way is to take significant wave height into account when developing wind GMF, as described by Lin and co-workers [34]. Nevertheless, this issue is beyond the scope of the text. The purpose of this research was to propose a method that can use high-resolution GNSS-R DDMs reconstructed from downscaled DDMs for wind speed retrieval. The performance of SR-DDM-derived wind speed should be as similar as possible to FR-DDM-derived wind speed. The results showed that

the proposed method reduces a considerable amount of data but only sacrifice a little performance. It is also believed that the proposed method can be applied to other GNSS-R applications, not limited to ocean wind sensing.

## 4. Discussion

The study presented here investigated an alternative way to retrieve the ocean wind speed from LR-DDMs. It was assumed that the LR-DDM, generated from the actual CYGNSS FR-DDM, is the first output product from the GNSS-R DDMI. The SR-DDM was then obtained by reconstructing the LR-DDM using the trained VDSR network. The retrieved wind speed derived from the SR-DDM shows consistent results to those of the FR-DDM.

Compared to general images (such as photos of landscapes, animals, or people), the DDM is a relatively simple image that makes the VDSR perform well for reconstruction. In addition, the trained DDM reconstruction network is a dedicated network. A concise VDSR network can provide the same super-resolution performance as an ordinary VDSR network because the network depth is adequate to store all of the features in the DDM. Although most of the training DDM came from the same satellite, the consistent reconstruction performance among different satellites was also attributed to the simplicity of the DDM.

In Section 3.1, we evaluated the reconstruction performance from the perspective of image processing. Several examples were selected to show the reconstruction results. It is apparent that both the SR-DDM and LSR-DDM, reconstructed from the LR-DDM, show extremely high similarities to the FR-DDM. Additionally, we also used two regular performance metrics (i.e., PSNR and SSIM) to verify the reconstruction results. According to the calculated PSNR and SSIM using the test data set, the reconstruction results can reach the image restoration standard and are consistent across different spacecraft. Furthermore, the results also show that the proposed method is insensitive to different scattering conditions. Although the reconstruction capability appears to increase with higher DDM SNR, the proposed method still produces high-quality SR-DDM even in the case of lowest DDM SNR. However, this paper only discusses one possible manner to verify the reconstruction method. Since no related research has been done before, we used a process based on image restoration to conduct the assessment. Overall, the image reconstruction quality evaluation compared the shapes between FR-DDM and SR-DDM. Consequently, it is necessary to investigate the feasibility of the SR-DDM for GNSS-R remote sensing. In this paper, we examined the retrieval performance of ocean surface wind speed. In the GNSS-R field, it is known that the DDM powers at the different delays and Doppler bins are correlated [35,36]. In the future, the statistical difference between the FR-DDM and SR-DDM will be compared. Furthermore, the statistics of the DDM may be viewed as a metric to adjust the reconstruction network to make reconstruction better.

Since we can reconstruct the SR-DDM, which is very close to the FR-DDM, from the LR-DDM, it is believed that we can extract usable observations from the SR-DDM. The results also show that the relationship between the FR-DDM-derived observations and wind speed, as well as the relationship between the SR-DDM-derived observations and wind speed, are highly similar. As a result, the SR-DDM retrieved wind speed showed performance that is at the same level as that of the FR-DDM retrieved wind speed. Although the overall performance appeared weak, especially in high winds, it should be noted that no calibration was applied, and only a simple regression model was used. The observation saturation phenomena, which caused significant bias in terms of high wind estimation, can be addressed by considering the ocean state during wind generation in the regression model.

From the perspective of generating DDM data products, the proposed method makes it possible to save approximately 94% of the data ($20 \times 128 \rightarrow 10 \times 16$). In addition, the downlink data were also reduced by almost 15% ($11 \times 17 \rightarrow 10 \times 16$). The overall performance degradation due to the inevitable super-resolution reconstruction distortion was only 0.007 m/s in terms of accuracy and 0.07 m/s in terms of precision for the worst case.

To implement the proposed method for a GNSS-R space mission, the instrument to process received reflected signals and produce DDMs should be modified first. For an existing GNSS-R receiver, this could be easily implemented by adjusting the correlator to multiply the received reflected signals with local code replica and carrier. The first output will be LR-DDM. Subsequently, the designer can choose to perform DDM reconstruction on board or at the ground station. In the first case, the designer can use the light reconstruction model to save power consumption. In the second case, the spacecraft only generates and collects the LR-DDM. The DDM reconstruction and wind speed retrieval can be conducted on the ground station after the LR-DDM is downlinked. However, regardless of the strategy, the FR-DDM recording mode still needs to be executed regularly. The purpose of the FR-DDM recording mode is to keep training the DDM reconstruction model. If the DDM reconstruction is conducted on board, then it is necessary to uplink the updated network parameters to the spacecraft.

## 5. Conclusions

In this paper, a preprocessing GNSS-R method for obtaining ocean surface wind speed based on the SR-DDM was proposed and validated using real CYGNSS scientific data products. We adopted the SISR algorithm and the VDSR network to implement the proposed method. The usage of the proposed method, as well as the preprocessing for the DDM, were provided. The approach was employed by utilizing 2018 globally distributed data products collected from the CYGNSS archive server along with interpolated ECMWF reference wind speed data on a randomly cross-validated train/test split data set. The advantages of the proposed method are the shrinking data size on DDM generation and, especially for space missions, the reduced data size on data downlink. However, the effect of reduced DDM size on wind speed retrieval is of great concern. Through the experiment using spaceborne data, the promising results can be summed up as follows. The retrieved wind speed using the SR-DDM ($20 \times 128$) reconstructed from the LR-DDM ($10 \times 16$) yielded a bias of 0.166 m/s and an RMSE of 2.134 m/s compared to the results based on the FR-DDM ($20 \times 128$) with a bias of 0.159 m/s and an RMSE of 2.064 m/s. In conclusion, the results indicated that the proposed method could save 94% of the data on DDM generation per channel with negligible wind retrieval performance degradation.

**Author Contributions:** Formal Analysis, H.-Y.W.; Project Administration, J.-C.J.; Supervision, J.-C.J. All authors have read and agreed to the published version of the manuscript.

**Funding:** This research was funded by the National Space Organization, Taiwan, grant number NSPO-S-107151.

**Acknowledgments:** The authors would like to thank the CYGNSS team for providing the scientific data used in this paper.

**Conflicts of Interest:** The authors declare no conflicts of interest.

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
