# Peer review of "Retrieval of Ocean Wind Speed Using Super-Resolution Delay-Doppler Maps"

_remotesensing, doi:10.3390/rs12060916_

Round 1

Reviewer 1 Report

see attached document

Reviewer 2 Report

General comments:

The authors applied the super-resolution algorithms on construct DDM. The algorithm shows peer performance compared with the conventional method, with smaller data volume. This is a contribution of knowledge. The language and paper management is good.

Detailed comments:

Section 2. Suggest to put the summary/overview of the proposed method at the front of the section, then introduce each subsection consequently. Page 5. “Among these methods, very deep super-resolution (VDSR) proposed by Kim and two other authors in 2016 is the first very deep model to perform SISR [9].” Explain the performance of the algorithms in the paper and why this algorithm is suitable for the problem is this paper explicitly. Page 7, Equation 5. The unit of the SNR is missing. Figure 7, the meaning of red lines, gray lines and dotted red line should be explained. Page 8-9, Section 2.5. Add more explanations in the paragraph. Page 10. Explain SSIM with more details. More references are required and explain the reason of selecting this metric. Page 12. You should emphasize that the Figure 10 and 11 use the SAME training data set. Page 13. One of the advantages of the proposed algorithm is the shrunken data size. You should discuss more on it to justify your statement and talk about its implementation. “Promising results were obtained.” How promising? Be precise.

Reviewer 3 Report

Retrieval of Ocean Wind Speed using Super-Resolution Delay-Doppler Maps

Wang and Juang

Summary:

Authors’ propose a method of analyzing CYGNSS data that results in a 4% degradation of resulting winds compared to using the full resolution data, while providing a 15% reduction in data needed to be downloaded from the satellite.  There is some question as to whether any degradation is tolerable.  However, this result may be of interest to those designing these missions (or other similar missions) and need to reduced data volume transmissions.  Overall, the paper is well written and well organized, although there are areas which need clarification or further explanation.  Substantively, I urge the authors to attempt a better high wind speed retrieval since this is the main aim of the CYGNSS satellite, to capture wind speeds below tropical storms and cyclones.  If this is beyond the scope of the text, perhaps the authors could provide more information as to why?  In any event, I recommend publication with minor revision.

Comments:

Introduction, p. 1: “a long integration period” is mentioned.  How long?

Introduction, p. 2: “the target could be easily recognized” is noted.  What target?  The winds or some physical target like an island in view?  “Target identification” is also mentioned a few lines down.  Please be more clear on what is meant by target.  Sounds like land areas?  

Figure 1: Suggest adding a description of what the red dots represent and the black rectangles.  I assume the dots are the LR-DDM sampling and the rectangles represent the FR-DDM grid.  It would be of interest to also add the C-DDM sampling if possible.

Table 1: I believe there is a typo in the Parameter name row.  Should the third column be C-DDM?  In the second row, should the dimension of the LR-DDM be 10x16 as stated in the text on p. 2?

p. 5: The statement “ The color represents the interpolated wind measurements processed from the ECMWF reanalysis products.” is not clear.  The color of the inset represents something different from the color of the main panel.  In the sentence before this one the authors refer to the main panel.  Thus the color represents the counts, not the wind speed itself.  Please correct or rephrase the sentence.

p. 5: “150,000 units” - where does this come from?  Are counts x 1000 in Figure 3?  

Figure 3: Are the ECMWF wind speed data daily averages along the satellite tracks or instantaneous? Please clarify the time period for each of the FR-DDM L0 data listed in Table 2.

p. 5: What do the authors mean by data #7 and #8 account for 70% of entire data set?  Can the authors please clarify what each of these data numbers represents?  Are they swaths?  Did #7 and #8 just contain more swaths?  Perhaps the authors should take more time in Section 2.1 to explain how the maps relate to the data granule numbers in Table 2.

p. 8: The discussion of Fig. 6b should mention not only the RMSE and bias but also the fact that the derived wind speeds are not linearly related to the ECMWF wind speeds.  High wind speeds are not captured by the GNSS-R data set.  Can the authors comment where they think the error lies at high wind speeds?  - I see that on p. 13 in the Discussion section the authors note the high wind speed issue.  Is it possible for the authors to attempt a high wind speed retrieval following the suggestions on p 13?  It seems to me that CYGNSS data are mainly for capturing high wind speeds under tropical systems so it is the high wind speed regime that is most interesting for your algorithm to be tested on.  

p. 10: Can the authors define “speckle” noise?

p. 13: Discussion: I am surprised that the authors do not compare their method to that of CYGNSS itself?  Does CYGNSS not provide their own wind speed estimates?  Have I missed something?  Is one of your comparisons equivalent to the C-DDM wind speed estimates?  Please clarify.

Reviewer 4 Report

Comments of the Manuscript “Retrieval of Ocean Wind Speed using Super-Resolution DelayDoppler Maps” by H.Y. Wang and J.C. Juang

This paper presents a new DDM reconstructing method for GNSS-R wind speed retrieval, which is verified with the CYGNSS DDM for wind speed retrieval. In general it's well writen, the methods are well described and the results presented in a logical, convincing manner. Some exceptions that, when addressed, will make the manuscript more clear, are given below.

Please note that I was not able to find any indicator of line number in the submitted manuscript, which makes it difficult to provide the comments with clear page and line numbers. I would suggest the authors could include the line numbers in the next submission.

1) Page 1: "... utilized to retrieve geophysical parameters such as ocean surface wind, vegetation, and soil moisture ...": please add relevant references for these applications. In addition, GNSS-R can be also used for altimetry (as one of the main applications), it would be helpful if the authors can put this application here and add corresponding references. "... mainly based on delay-Doppler maps (DDMs) ...": it's right that most of GNSS-R applications use DDM as the main observations. However, DDM can only provide the power information of the reflected GNSS signal. There are some other applications using the carrier phase information. It would be better if the author can put a sentence for clarification, e.g., "It is noted that the several researchers have been performing GNSS-R studies with phase observation (with relevant references), which is not included in DDMs and is not discussed in this paper".

2) Introduction, the 2nd paragraph: the authors introduce many references on NRCS reconstruction from the GNSS-R DDM. I am not sure if these references are relevant to the method proposed in this paper. If I understand correctly, these references are mainly on the transformation from delay-Doppler domain observations to spatial domain NRCS. Do you want to claim that the high resolution DDM can benifit these reconstructions?

3) Table 1: I found two "SR-DDM" in the first row of the table. The first "SR-DDM" should be "C-DDM", right?

4) Major -- Section 2.3 Supper-resolution DDM reconstruction Model: it is understood that the SR-DDM is reconstructed with a fancy deep learning method. Is it possible to use some simpler methods, e.g., direct 2-D interpolation to reconstruct the SR-DDM? What would be the expected performance of these simpler methods if it is possible?

5) Section 3.1: it is known that the DDM powers at different delay and Doppler bins are correlated (10.1109/TGRS.2016.2579504, 10.1109/TGRS.2017.2785343). In addition to the shape of the FL- and SR-DDMs, is it possible to check the statistics of these DDMs? If it is to difficult to compute these statistics with the current CYGNSS data, then it is something to consider as future work.

6) Major -- Section 3.2: the wind speed retrieval algorithm is based on the SNR of the DDM computed around its peak. If we only need the information around the DDM peak, why do we have to downlink LR-DDM instead of the C-DDM (as they have similar size)? In addition, I'm not sure if the consistency of the peak power between the FR-DDM and SR-DDM is sufficient to verify the DDM reconstruction methods. It would be helpful if the authors could think about that.

Round 2

Reviewer 1 Report

see attached document

Reviewer 3 Report

Summary:

The authors have mostly clarified the issues I had with the original text.  Overall, I have no further significant corrections, only a few minor edits I found while reading through the text, aside from the issue with Fig 3a and associated text at line 143, which I think needs to be addressed.  I recommend publication at this point and do not need to see the manuscript again.

Minor edits:

line 80: Needs revising, sentence does not make sense.  Maybe a typo?

line 122: Space needed after "date,".

Lines 143-144: This sentence needs revising.  It is grammatically incorrect and thus difficult to understand as written.  I believe "amount" can just be deleted to get the intended meaning, but I am not sure.  And again, I think I noted this in the first version, the 150,000 counts seems orders of magnitude above the scale in Fig 3a.  The authors stated they corrected this issue but it is still not clear what the 150,000 is referring to.  Should it be 150?  Or are you talking about the sum of all the points within the region of high counts?  I see that Table 2 shows 150,000 counts from #7 and #8, but were all of these counts off the east coast of the US?  If so, why is the color bar only going to 120?

Figures: The quality of the images will need to be better for press.  They are very blurry.

Reviewer 4 Report

please see my comments in attached document.

Round 3

Reviewer 1 Report

This reviewer congratulates the authors for this nice work.